# Programming and Setting Up the Object Detection Algorithm YOLO to Determine Feeding Activities of Beef Cattle: A Comparison between YOLOv8m and YOLOv10m

**DOI:** 10.3390/ani14192821

**Published:** 2024-09-30

**Authors:** Pablo Guarnido-Lopez, John-Fredy Ramirez-Agudelo, Emmanuel Denimal, Mohammed Benaouda

**Affiliations:** 1Institut Agro Dijon, 26 bd Docteur Petitjean, 21079 Dijon, France; pabloguarnido@hotmail.com (P.G.-L.); emmanuel.denimal@agrosupdijon.fr (E.D.); 2Grupo de Investigación en Ciencias Agrarias—GRICA, Escuela de Producción Animal, Facultad de Ciencias Agrarias, Universidad de Antioquia, Medellin 050010, Colombia; johnf.ramirez@udea.edu.co

**Keywords:** computer vision, feeding activities, beef cattle, YOLO, precision livestock farming

## Abstract

**Simple Summary:**

This study addresses the challenge of accurately monitoring the feeding behavior of cattle, which is crucial for their health and productivity. The aim was to compare two versions of a computer vision algorithm, YOLO (v8 vs. v10), which identifies objects in images, to evaluate how well they can recognize the feeding activities of beef cattle. By recording videos of bulls on a farm and analyzing them using YOLO algorithms, we found that both versions were effective at detecting these behaviors, but the latest version was slightly better and faster at learning. This new version also showed a reduced tendency to repeat errors. The conclusion is that the latest version of YOLO is more efficient and reliable for real-world use on farms. This advancement is valuable to society as it helps farmers better monitor and manage cattle feeding, leading to healthier animals and more efficient farming practices.

**Abstract:**

This study highlights the importance of monitoring cattle feeding behavior using the YOLO algorithm for object detection. Videos of six Charolais bulls were recorded on a French farm, and three feeding behaviors (biting, chewing, visiting) were identified and labeled using Roboflow. YOLOv8 and YOLOv10 were compared for their performance in detecting these behaviors. YOLOv10 outperformed YOLOv8 with slightly higher precision, recall, mAP50, and mAP50-95 scores. Although both algorithms demonstrated similar overall accuracy (around 90%), YOLOv8 reached optimal training faster and exhibited less overfitting. Confusion matrices indicated similar patterns of prediction errors for both versions, but YOLOv10 showed better consistency. This study concludes that while both YOLOv8 and YOLOv10 are effective in detecting cattle feeding behaviors, YOLOv10 exhibited superior average performance, learning rate, and speed, making it more suitable for practical field applications.

## 1. Introduction

In recent years, scientific interest in analyzing animal behavior as a cornerstone for informed decision-making in cattle farming has increased. Accurate recording of parameters such as feeding frequency and duration provides invaluable insights into the nutritional monitoring of cattle, facilitating the formulation of balanced diets that ensure well-being, productivity, and a reduction in the environmental impact of the herd [1]. Furthermore, beyond nutritional considerations, deviations in feeding behavior patterns, such as reduced intake or abnormal feeding habits, can serve as early indicators of underlying health issues [2]. However, continuous visual observation is labor-intensive, time-consuming, and not worth the limited benefits gained. Since the emergence of Industry 4.0 technologies in the livestock sector, machine learning algorithms coupled with cameras have assisted in this task over the past decades. These machine learning algorithms, specifically object detection algorithms, make it feasible and efficient to assess individual animal behaviors across diverse farm sizes and types, showcasing their versatility and applicability across various livestock management contexts [3].

When discussing object detection (involving many objects in a single image), the You Only Look Once (YOLO) algorithm has already demonstrated its utility in monitoring feeding and animal behavior across different species [4,5,6,7]. Regarding cattle, YOLO has been used to detect feeding behaviors in cows [8,9], monitor estrus [10], and track individual cattle behavior [11]. While other computer vision algorithms, such as ResNet, Faster R-CNN, and RetinaNet, have been applied in animal science, YOLO represents a favorable balance between accuracy, a unified structure, flexibility, and, crucially, when considering on-farm applications, high speed and real-time performance [12]. Several authors have compared YOLO’s performance in the real-time object detection of cattle and emphasized its potential for monitoring multiple animals simultaneously in various feeding environments [13].

The YOLO algorithm was created through DarkNet and was first presented in June 2016 at the Caesar’s Palace Conference Center in Las Vegas, Nevada, by Joseph Redmon [12]. Over the following years, he published improved versions of the algorithm—YOLOv2 [14] and YOLOv3 [15]—until he stopped his research career due to concerns about the military applications of his algorithm. Bochkovskiy continued Redmon’s work, releasing YOLOv4 [16]. Two months after YOLOv4’s launch, Glenn Jocher from Ultralytics^®^ released YOLOv5, which was developed using PyTorch instead of DarkNet [17]. After this version, the YOLO algorithm continued to be developed as open-source by independent programmers, leading to the most recent version, YOLOv10. In this work, we focused on YOLOv8, the latest version significantly enhanced by Ultralytics^®^, and YOLOv10 (or YOLOX), the most recent version developed by independent researchers (THU-MIG, Tsinghua University’s Multimedia Intelligence Group). Real-time detection algorithms are among the tools shaping the future of technologies used in animal production due to their ability to provide solutions that aid in decision-making on farms. Therefore, we concentrated on using the YOLO algorithm in this study to determine feeding behavior in cattle.

Other studies have evaluated cattle behavior from a flank view [18] or a top view [19]. In this study, we recorded feeding behavior from the front view to better capture the entire head movement of the animal and the close interaction between the animal’s mouth and the feed. Specifically, this work focused on three distinct activities at the feeder: (1) visiting, which indicates the animal’s presence without ingesting; (2) chewing, which reflects the animal’s health status and rumination function; and (3) biting, which corresponds to the act of eating itself, allowing the determination of ingestion. These activities are the most representative feeding behaviors of cattle and allow for the estimation of other relevant performance indicators such as individual intake, ingestion time, eating rate, and health status [2]. The objective of this study was to compare the performance of YOLOv8 and YOLOv10 models in detecting the following three key activities during the feeding behavior of steers: biting, chewing, and visiting the feeder.

## 2. Materials and Methods

### 2.1. Animals, Diet and Measurements

Videos were recorded on a commercial Charolais farm covering 173 hectares (Tart-Le-Bas, Burgundy, France), which is located at the agricultural high school of Quetigny, France. For this experiment, a total of 12 young Charolais bulls (581 ± 37 kg, 16 ± 1.4 months old) were monitored for individual dry matter intake (DMI). In France, commercial beef fattening usually takes place on former dairy farms; therefore, the feeders are similar to those found on dairy farms, which influenced the decision to place the camera in front of the animals. Animals were recorded for 7 min per day just after feed distribution, with one video recorded per day, yielding a total of 24 videos over 24 non-consecutive days across two consecutive months. Videos averaged 7 min because, after this time, most animals left the feeder. In addition, the video storage capacity was also a limiting factor. The animals were housed in a covered barn with straw bedding and were fed twice daily: first at 8:00 AM with alfalfa hay ad libitum, as well as an energy and protein concentrate, and again at 4:00 PM with just alfalfa hay [90% DM, 0.7 UFV, 483 g NDF/kg DM, 175 g CP/kg DM] ad libitum. Video recordings of intake and ingestion time were conducted during the second meal distribution. The reason for recording animals during the second meal was that during the first meal, the animals were hungrier, and dominance behaviors that could affect prediction were more likely to occur. Fresh matter intake was measured by manually weighing individual feed amounts per animal [offered feed minus refused feed] using an electronic scale (Rubbermaid^®^ Digital Utility Scale–400 lbs ×.5 lb). Samples to measure DM were taken weekly, stored in a homogeneous manner, and analyzed in an external laboratory.

### 2.2. Recording System

Videos were recorded using an RGB-D camera, Intel^®^ RealSense™ D455 (Intel, Santa Clara, CA, USA), mounted on a tripod and connected to a computer, as shown in Figure 1. The videos were captured using Intel software (Intel RealSense SDK 2.0 v2.51.1). During the recordings, animals were manually identified by their number to enable individual predictions later. The camera used in this study has three sensors—an infrared sensor, an RGB sensor, and a depth sensor (3D). The camera has an RGB depth field of view of 90° × 65°. In this study, only two dimensions were required, so only the RGB sensor was used. The camera software was configured to record videos at 5 frames per second. To minimize the impact of lighting bias, videos were recorded at the same time each day, from the same angle, and at a distance of 5 m from the center of the feeder.

### 2.3. Data Set Description and Labelling

Intel software was used to extract individual frames from the videos. Fifty frames per video were randomly selected to create a database (1200 frames in total), which was divided into three datasets—a training set (70%), a validation set (10%), and a test set (20%). During the study period, 20 videos were recorded, numbered sequentially from 1 to 20 to reflect the order in which they were captured, documenting cattle growth and changes in environmental conditions. To minimize biases and ensure balanced representation in our datasets, we strategically divided the videos based on their numbering: odd-numbered videos were used for the training dataset, while even-numbered videos were designated for the test dataset. This method ensured that both datasets included varied images throughout the entire period, maintaining a strict separation between training and test data to accurately assess the model’s ability to generalize to new, unseen conditions. Figure 2 shows examples of the image diversity used in this work.

The frame subsets were manually labeled using the online software “Roboflow” “https://roboflow.com/” (accessed on 12 September 2023). Roboflow (Figure 3) enables users to generate the necessary YOLO text files for training and evaluation. These files contain annotations in a specific format that includes the class label and normalized coordinates of the bounding boxes representing the object’s location in the image. Each line in a text file corresponds to one object and follows this format: <object-class> <x_center> <y_center> <width> <height>, where all values are normalized between 0 and 1 (e.g., 0 0.534 0.622 0.142 0.256) [20].

Three distinct feeding behaviors were meticulously identified and labeled in the selected frames, as depicted in Figure 4. These behaviors were classified as follows:Visiting: Characterized by the animal standing with its head elevated and not engaging in any feeding activity, signifying the absence of feed intake.Biting: Defined by the animal lowering its head toward the feeder, suggesting active engagement with the feed and typically indicating the initial action of feed intake.Chewing: Marked by the animal raising its head yet displaying clear signs of mastication, evidenced by the presence of feed in the mouth.

Figure 4 provides visual examples of these behaviors, each captured from a front-facing perspective to ensure clarity in the observable actions. The images serve as a visual reference for the classification criteria applied during the manual labeling process, enabling precise and consistent categorization across the datasets.

### 2.4. YOLOv8 and v10 Network Structure

The YOLOv8 and v10 algorithms are based on the same principles but have different neural network structures (Figure 5). These models represent significant updates in the YOLO (You Only Look Once) series, known for their real-time object detection capabilities. Each model seeks to push the boundaries of speed, accuracy, and efficiency in object detection. YOLOv8, building upon advancements from previous iterations, introduces several architectural improvements aimed at enhancing model performance and efficiency. It continues to leverage components like CSPNet from earlier versions but also incorporates new methods to optimize latency and parameter efficiency. The shift from a traditional backbone like CSP-Darknet53 to more efficient designs enables YOLOv8 to offer better performance with reduced computational overhead [21,22,23,24]. YOLOv10, the latest in the series, brings even more profound architectural innovations, focusing on both model efficiency and accuracy. One of the key innovations is the introduction of a lightweight classification head that utilizes depth-wise separable convolutions—a technique that separates the convolutional process into depth-wise and point-wise operations. This adjustment significantly lowers computational costs and reduces model parameters without sacrificing performance. Additionally, YOLOv10 incorporates holistic model design strategies, such as the consistent dual assignments for NMS-free training and rank-guided block design, further enhancing its efficiency and effectiveness. Extensive testing shows that YOLOv10 provides state-of-the-art performance and efficiency across various model scales, demonstrating improvements in both average precision and inference latency compared with its predecessors [25].

### 2.5. Training

The models were trained on Google Colab utilizing a Tesla T4 GPU with 15,360 MiB of memory. Necessary libraries, such as “numpy” for numerical operations, “cv2” for image processing, and the YOLO models from the “ultralytics” package, were imported. Additionally, “supervision” and “roboflow” libraries were installed to assist with model training and data handling. The YOLO models were initialized with pre-trained weights. These weights serve as a starting point, allowing the model to build upon previously learned features, thereby speeding up the training process and improving the initial performance. The dataset configuration file (“data.yaml”) specifies the training and validation data paths as well as the number of classes. This file is essential for informing the model about the structure and content of the dataset. The training command was issued using the “yolo” command-line interface. Key parameters include the following: Task and Mode—the task was set to object detection (“detect”), and the mode was set to training (“train”); Model and Data—the model was specified, and the dataset configuration file was provided (“data.yaml”); Training Parameters—the models were trained for 500 epochs with an image size of 640 pixels and a batch size of 8. These parameters control the duration and intensity of the training process; Patience—the “patience” parameter was set to 50, meaning that if validation performance did not improve for 50 consecutive epochs, training would stop early to prevent overfitting.

During training, the model used automatic mixed precision (AMP) to speed up computation and reduce memory usage. The model architecture, including layers and parameters, was printed for verification. Data augmentation techniques, such as blur and color adjustments, were applied to the training images to improve the model’s robustness. The optimizer used for training was “AdamW”, which was automatically selected to optimize the learning rate and momentum parameters. The model logged its progress to TensorBoard, allowing for the real-time monitoring of training metrics, such as loss and accuracy. Throughout the training process, the model periodically validated its performance on the validation dataset. This validation helped monitor the model’s ability to generalize to new data and prevent overfitting. The training continued for the specified number of epochs or until early stopping criteria were met. Upon completion, the model’s weights were saved.

### 2.6. Evaluation Indicators

To accurately evaluate the performance of the models, we used common evaluation indicators in target detection algorithms: precision, recall, mean average precision (mAP), and F1-score. In terms of precision and recall, there are four possible outcomes when predicting a test sample: True Positive (TP), False Positive (FP), True Negative (TN), and False Negative (FN). These evaluation indicators are defined as follows:Precision is the ratio of TP predictions to the total number of positive predictions made by the model (both TP and FP). It reflects the accuracy of the positive predictions.Recall is the ratio of TP predictions to the total number of actual positive cases (TP and FN). It measures the model’s ability to identify all relevant instances.Average Precision (AP) is defined as the area under the precision-recall curve; AP provides a single value that summarizes the model’s precision and recall performance at various threshold levels.Mean Average Precision (mAP) is the mean of the average precision values for all classes. It serves as a comprehensive measure that evaluates the overall performance of the model across different object classes.F1-Score is the harmonic mean of precision and recall. It balances these two metrics by providing a single score that accounts for both false positives and false negatives.

Additionally, the changing trend of the model’s loss curve can also be used to assess the model’s performance. A faster loss curve fitting speed, better fit, and lower final loss value generally indicate stronger performance. Furthermore, a Python code was developed to evaluate the performance of the trained object detection models using a set of test images and their corresponding annotations. The process begins by importing necessary libraries for numerical operations, image processing, file handling, and model operations. The Intersection over Union (IoU) function is defined to calculate the overlap between predicted and ground-truth bounding boxes, providing a measure of prediction accuracy. The code reads the ground truth annotations from the test dataset, which are formatted in YOLO style and converted into absolute coordinates. The trained YOLO model is then loaded using the specified model weights and directories for test images, and their annotations are set. The code initializes dictionaries to count TP, FP, and FN for each class and sets up lists to store precision and recall values. The code iterates through each image in the test directory, reading the image and its corresponding ground truth annotations. The model makes predictions, extracting bounding boxes and their corresponding class labels, which are then compared with the ground truth annotations. If a prediction matches a ground truth (having the same class ID and an IoU greater than 0.5), it is counted as a TP. If no match is found, the ground truth is counted as an FN, and any remaining predictions are counted as FP. After processing all images, the code calculates precision, recall, F1-score, and average precision for each class.

## 3. Results

### 3.1. YOLOv8 and v10 Performance in Feeding Behavior Detection

Table 1 highlights that YOLOv10 generally outperforms YOLOv8 across several metrics. For instance, YOLOv10 shows a higher mean Average Precision (mAP) of 0.94 compared to 0.92 for YOLOv8, indicating an overall improvement in object detection performance. For the “biting” activity, both models exhibit excellent performance with nearly perfect precision, recall, and F1-scores. However, for the “chewing” activity, YOLOv10 demonstrates higher precision, recall, and F1-score than YOLOv8, signifying better detection accuracy and reliability. In the “visiting” activity, YOLOv8 achieves perfect precision but significantly low recall, resulting in a low F1-score. In contrast, YOLOv10 presents a more balanced performance with considerably improved recall and F1-score, though with a slight decrease in precision.

On average, the metrics for the “visiting” activity are significantly lower than those observed for “chewing” and “biting”. This discrepancy may be due to the following two factors: (1) the lower number of instances of “visiting” compared to the other activities, which impacts the model’s training and thus the accuracy for this activity, and (2) the “visiting” activity is more ambiguous as it only relates to the presence of the animal without any feeding behavior (chewing or biting), making it more difficult to define. These metrics collectively suggest that YOLOv10 offers more robust and reliable performance across different activities, making it a superior choice for applications requiring high-accuracy object detection in our database. The number of instances differs between models because YOLOv10 did not detect some instances that YOLOv8 did, leading to a lower count of instances for certain classes in the YOLOv10 evaluation. This discrepancy arises due to the models’ differing abilities to detect objects with an Intersection over Union (IoU) greater than 0.5 and correctly match the activity labels.

Figure 6 shows an example of the results of the animals’ feeding behavior recorded through a frontal view with the predicted result (by the YOLO algorithm) of individual feeding behavior. As can be seen from Figure 6 and in accordance with results shown in Table 1, both versions of the YOLO algorithm can accurately identify animals’ ‘Biting’ and ‘Chewing’ activities with a confidence level above 0.98.

### 3.2. Confusion Matrix of Feeding Activities Predicted with YOLOv8 vs. v10

Figure 7 displays the normalized confusion matrices for YOLOv8m and YOLOv10m. Both models demonstrate excellent performance in accurately predicting ‘Biting’ and ‘Chewing’ behaviors, with YOLOv8m achieving 0.98 accuracy for both and YOLOv10m achieving 0.98 and 0.99, respectively. However, both models exhibit a tendency to confuse ‘Visiting’ with ‘Chewing’. Notably, YOLOv8m shows greater confusion in this regard, with only 0.15 accuracy in correctly identifying ‘Visiting’ compared to 0.37 accuracy observed in YOLOv10m. This indicates that while both algorithms are highly effective at recognizing ‘Biting’ and ‘Chewing,’ YOLOv10m, despite its overall precision, struggles more with distinguishing ‘Visiting’ from ‘Chewing.’ This confusion can be explained by the similarities between these two activities and the relatively few instances of ‘Visiting’ recorded in the database.

### 3.3. Learning Rates and Parameters of YOLOv8 and v10

The learning rates of both YOLOv8 and YOLOv10 are shown in Figure 8. The comparative analysis reveals that while both YOLOv8 and YOLOv10 models are effective, YOLOv10 generally exhibits better stability and lower validation losses across various metrics. YOLOv8, on the other hand, converges faster during training but shows higher validation losses, indicating potential overfitting. The consistently lower validation losses of YOLOv10 suggest better generalization and robustness when applied to unseen data.

Finally, to achieve the best performance in predicting feeding activities, we set up YOLOv8 and YOLOv10 with the standard parameters shown in Table 2.

## 4. Discussion

As computer vision continues to grow in prominence within livestock management, it is essential to evaluate the most commonly used algorithms. In this context, the present study assessed two versions of object detection algorithms, YOLOv8 and YOLOv10, which represent different development approaches within the YOLO framework. YOLOv8 is an improved version of the original YOLO structure created by its inventor, while YOLOv10 was developed by independent researchers. The objective was to evaluate the performance of these two versions in predicting cattle feeding behaviors, which are critical to cattle productivity, health status, and daily performance. On average, YOLOv10 demonstrated slightly better accuracy than YOLOv8 in predicting feeding activities and distinguishing between them (confusion matrix). Moreover, YOLOv10 showed improved learning rate outcomes, suggesting better overall model performance.

We evaluated the performance of YOLOv8 and YOLOv10 using several key metrics: precision, recall, mean Average Precision (mAP), and F1-score. Precision, which measures the accuracy of positive predictions, was particularly high for both models in detecting the “biting” activity (0.99 for both YOLOv8 and YOLOv10). This indicates a strong capability to correctly identify this behavior without false positives. Recall, which assesses the model’s ability to identify all relevant instances, was lower for the “visiting” activity, particularly in YOLOv8 (0.15). This suggests that the model had difficulty detecting all instances of this behavior, potentially due to the fewer occurrences and the nature of the activity. The mAP metric, which provides a comprehensive measure of the model’s performance across different detection thresholds, was higher in YOLOv10 (0.94) than in YOLOv8 (0.92). The F1-score, a harmonic mean of precision and recall, further highlights the overall performance. For “chewing”, YOLOv10 outperformed YOLOv8 (0.93 vs. 0.91), indicating better detection reliability. The slight improvement in mAP and F1-score in YOLOv10 suggests it may be better suited for applications requiring high accuracy, especially in detecting less frequent behaviors like “visiting”.

However, YOLOv8’s faster convergence might make it a better choice in scenarios where training time is limited and high accuracy across all metrics is not as critical. A survey comparing YOLO versions from YOLOv1 to the state-of-the-art YOLOv10 has consistently shown that newer versions offer better performance metrics like precision and recall due to architectural refinements [29]. For instance, YOLOv10 integrates advanced post-processing techniques and anchor-free detection heads, which further reduce computational overhead while improving detection accuracy. YOLOv10 builds upon the advancements of its predecessors by optimizing both the architecture and post-processing stages, leading to superior performance in real-time object detection tasks [30]. Both biting and visiting activities were predicted with high precision and recall (>0.98) by both YOLO versions (Figure 9). This success is likely due to the distinct head movements associated with these activities: head down touching the feed (biting) versus head up with the mouth closed (visiting). However, the main challenge arose with the “chewing” activity, which had lower prediction performance and was often confused with “visiting” (especially by YOLOv8). This confusion can be explained by the subtle differences between these two activities—mouth closed (visiting) versus mouth open with feed present (chewing). Other studies have similarly pointed out the difficulty in determining chewing activity [31,32]. To address this issue, other authors have proposed the following methods: (1) using accelerometers to differentiate feeding activities based on head position [31]; (2) estimating chewing through sound analysis (or combining video and sound), which offers an interesting proxy by considering both visual and auditory differences [33,34,35]; and (3) incorporating multiple-frame tracking algorithms into YOLO, which may allow the algorithm to better capture jaw movements and improve prediction accuracy. This multi-frame algorithm has already been applied with YOLO [36,37], and future research could evaluate its efficacy in improving activity prediction in this context. This method could increase both YOLO’s prediction performance and reduce the confusion between activities.

YOLOv10’s superior performance in detecting the “visiting” activity, as evidenced by its higher recall and F1-score compared to YOLOv8, can be attributed to several key factors related to its architectural improvements and their impact on object detection capabilities. YOLOv10 incorporates a more refined network structure that includes a lightweight classification head with depth-wise separable convolutions [25]. This structural change reduces computational costs and enhances the model’s ability to generalize across different classes, particularly those with subtle distinctions, such as “visiting” versus other feeding behaviors. As highlighted by the developments in the YOLO series [25], including versions YOLOv6, YOLOv7, and YOLOv8, the architectural enhancements have significantly improved the models’ feature extraction and classification capabilities. These advancements include the introduction of decoupled head structures, enhanced neck modules for better feature aggregation, and advanced convolutional layers. These features are particularly crucial for tasks requiring fine-grained distinctions, as they enable the models to better capture and classify subtle details in the input data. While YOLOv10 shows robust overall performance, it is important to acknowledge areas where YOLOv8 exhibited strengths, particularly in faster convergence during training. YOLOv8 demonstrated quicker attainment of lower training and validation losses, which can be advantageous in scenarios where computational resources are limited or rapid model deployment is necessary. This quicker convergence suggests that YOLOv8 may be more efficient in learning from data early in the training process. However, YOLOv8’s performance comes with a trade-off. Despite its faster convergence, YOLOv8 may not generalize as well in detecting less frequent or more subtle behaviors, such as the “visiting” activity, where an animal is present at the feeder without actively feeding. This could lead to the underreporting of critical events related to animal monitoring. In contrast, YOLOv10, although requiring a longer training period and maintaining slightly higher losses, offers a more balanced performance across all activities [38]. This balance makes YOLOv10 more suitable for real-time applications in precision livestock farming, where accuracy and reliability are paramount.

Finally, regarding the slight differences in learning rate between YOLOv8 and YOLOv10, we recommend using the latter due to its improved processing speed and slightly greater capacity for deployment under real-world conditions [39]. However, we acknowledge that the differences between both versions are minor, and both versions (with their different model structures) may perform well in field conditions. The next step is to correlate these predicted activities with other parameters, such as health status, fertility, or performance traits.

## 5. Conclusions

In this paper, we summarized the evolution of YOLO, one of the most well-known object detection algorithms in computer vision, highlighting how YOLOv8 represents the last version closely following the original architecture, while YOLOv10 has been improved by independent developers, both as open source. We compared these two versions of the YOLO algorithm to predict feeding activities in cattle from a frontal view. These feeding activities are becoming increasingly relevant in animal production due to their relationship with health, animal performance, and efficiency. The results of this study show that both versions of the YOLO algorithm performed similarly in predicting feeding activities, in terms of accuracy (precision, recall, and mAP) and confusion matrix, with YOLOv10 slightly outperforming YOLOv8. However, where the difference is more pronounced between these two algorithms is in the speed at which they reach optimal training and performance, with YOLOv8 being superior to YOLOv10 (132 and 204 epochs, respectively). In conclusion, both YOLOv8 and YOLOv10 are suitable for predicting “biting” and “chewing” activities in beef cattle with an accuracy of around 98% using our database. In future research, these activities could be evaluated from different points of view. The automated detection of these feeding activities on farms may improve decision-making for cattle producers in terms of nutritional strategies, early detection of abnormal health status, or management routines adapted to animal behavior.

## Figures and Tables

**Figure 1 animals-14-02821-f001:**
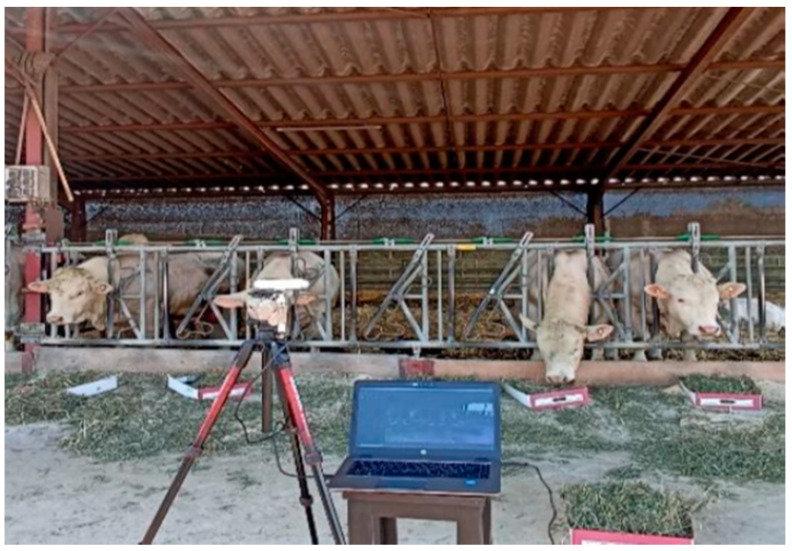
Recording set of animals and cameras.

**Figure 2 animals-14-02821-f002:**
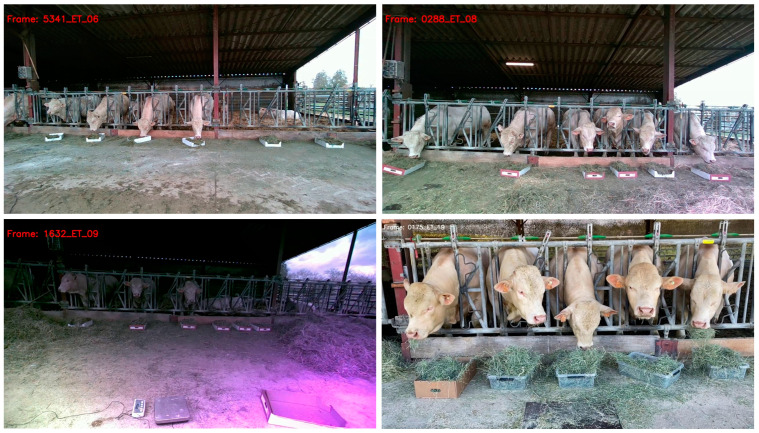
Examples of the images used in individual activities classification.

**Figure 3 animals-14-02821-f003:**
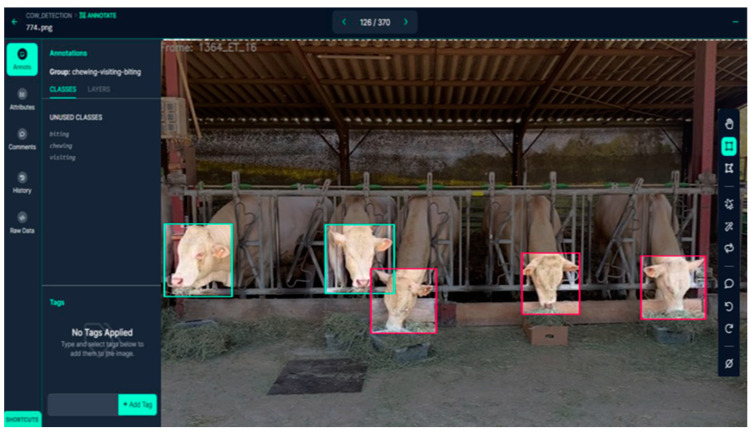
Roboflow software to label individual activities of cattle.

**Figure 4 animals-14-02821-f004:**
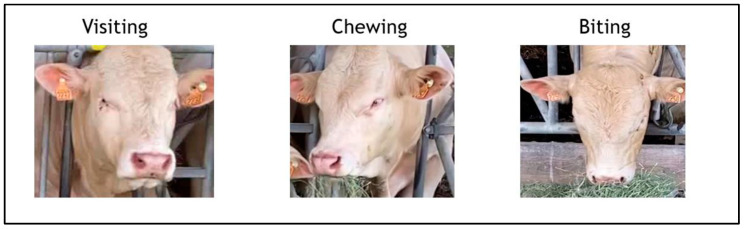
Three main feeding activities recorded and determined in beef cattle.

**Figure 5 animals-14-02821-f005:**
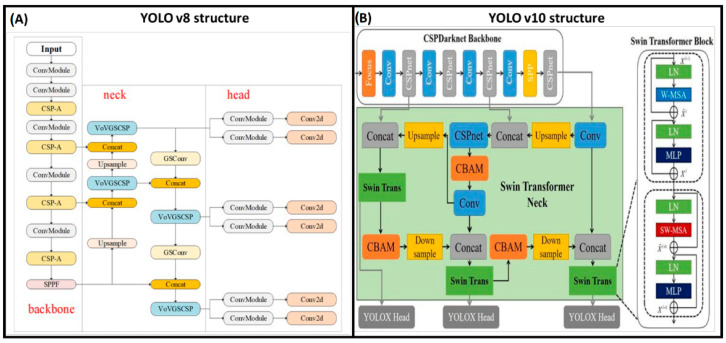
Differences in the neural network structure of YOLOv8 (**A**) and v10 (**A**). The image in panel A has been adapted from Shao et al. (2024) [26] and the image in (**B**) has been adapted from Xu et al. (2022) [27].

**Figure 6 animals-14-02821-f006:**
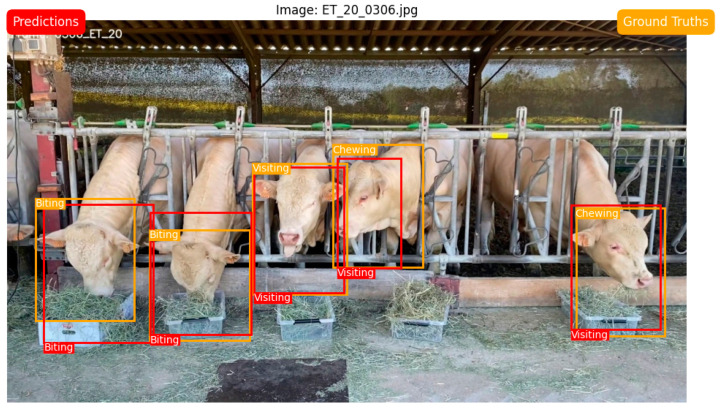
Capture of a frame showing the YOLOv8 prediction of feeding activities of individual beef cattle animals.

**Figure 7 animals-14-02821-f007:**
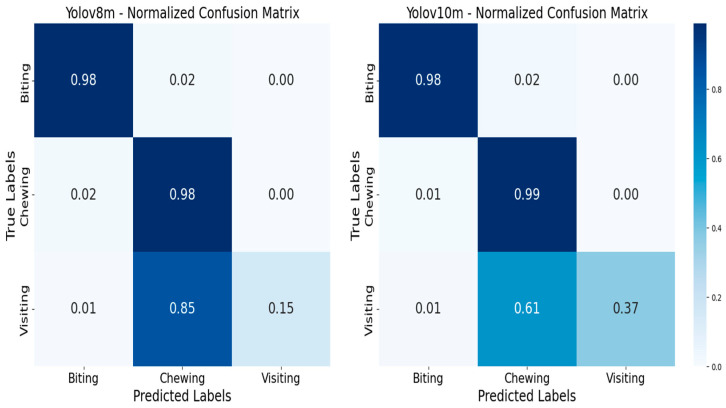
Confusion matrix of main activities analyzed by YOLOv8 vs. v10.

**Figure 8 animals-14-02821-f008:**
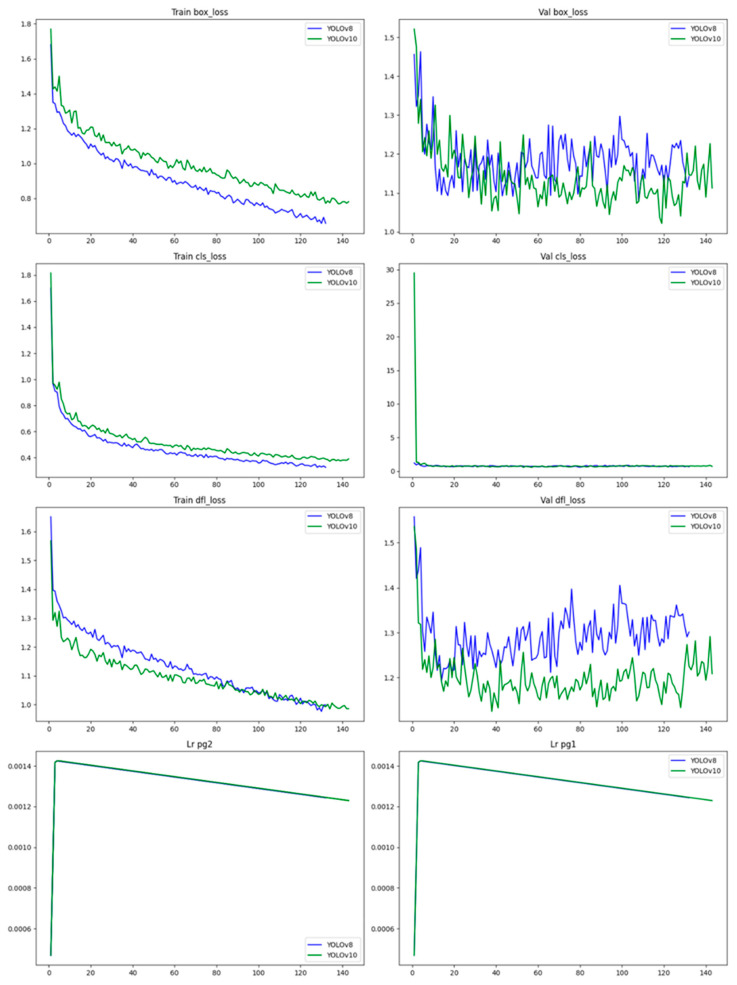
Learning rates of YOLOv8 and YOLOv10.

**Figure 9 animals-14-02821-f009:**
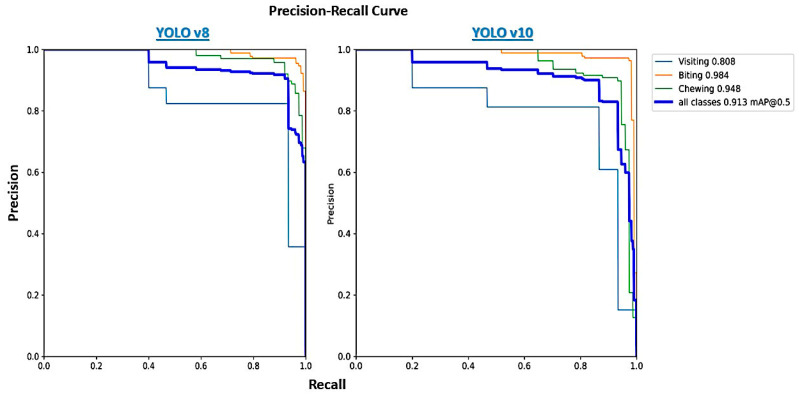
Precision–Recall curves of YOLOv8 and YOLOv10.

**Table 1 animals-14-02821-t001:** Results of YOLO model performance classifying feeding activities of cattle divided by version v8 vs. v10.

Model	Class	Instances ^µ^	Precision *	Recall *	F1-Score *	mAP *
YOLOv8	All	2040	-	-	-	0.92
	Biting	1128	0.99	0.98	0.99	-
	Chewing	762	0.84	0.98	0.91	-
	Visiting	150	1.00	0.15	0.26	-
YOLOv10	All	1953	-	-	-	0.94
	Biting	1081	0.99	0.98	0.99	-
	Chewing	737	0.87	0.99	0.93	-
	Visiting	135	0.98	0.37	0.54	-

^µ^ Number of instances is the number of times that one precise activity appears (one activity can be several times in the same frame). * Precision, recall, and mean average precision (mAP) reflect the model’s performance.

**Table 2 animals-14-02821-t002:** Standard parameters used in YOLOv8 and v10.

Feature *	YOLOv8	YOLOv10
Layers	295	498
GFLOPs	79.1	64.0
Optimizer	AdamW	AdamW
Learning Rate	0.01	0.01
Momentum	0.937	0.937
Weight Decay	0.0005	0.0005
Warmup Epochs	3.0	3.0
Training Epochs	1000	1000
Batch Size	8	8
Image Size	640	640
Freeze Layers	model.22.dfl.conv.weight	model.23.dfl.conv.weight
Augmentations	Blur, MedianBlur, ToGray, CLAHE	Blur, MedianBlur, ToGray, CLAHE
Mixed Precision	Yes	Yes
Max Detections	300	300
Classes	3	3
Patience	50	50

* To better understand these parameters, previous researchers have reviewed them, explaining their meaning and influence on model predictions [28].

## Data Availability

The data presented in this study are available on request from the corresponding author.

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
