# Peer review of "Programming and Setting Up the Object Detection Algorithm YOLO to Determine Feeding Activities of Beef Cattle: A Comparison between YOLOv8m and YOLOv10m"

_animals, 2024, doi:10.3390/ani14192821_

Round 1

Reviewer 1 Report

Comments and Suggestions for Authors

The main issue addressed by the research is the effectiveness of YOLO v8 and v10 algorithms in detecting feeding behaviors in cattle from frontal images. The study aims to evaluate which version of the algorithm performs better in identifying specific activities such as “Chewing,” “Biting,” and “Visiting.” Although the research tackles a relevant practical application, the choice of these specific algorithms and the rationale for comparing only these versions could be more thoroughly justified. An analysis explaining why these algorithms were chosen over more recent ones or alternative technological approaches would strengthen the study. The authors also need to better justify the significance of each behavior in the study and its impact.

The originality of the research lies in the practical application of YOLO algorithms for detecting specific feeding behaviors in cattle, a field that can significantly impact efficient herd management and health. The specific gap addressed is the lack of detailed comparison between the latest YOLO versions in the context of cattle monitoring. The study contributes by demonstrating how architectural improvements in YOLO v10 result in superior performance compared to YOLO v8. However, a deeper exploration of the relevance of these feeding behaviors to productivity and animal health, as well as a discussion on how automated detection can be integrated into existing management practices, would provide additional value.

This study offers valuable insights by directly comparing YOLO v8 and v10 in the specific task of detecting feeding behaviors in cattle. It provides a detailed analysis of architectural improvements and their practical implications. While many studies focus on general performance comparisons, this work highlights practical and technical differences between algorithm versions, which can guide more informed choices for automated monitoring tools. Nevertheless, the study would benefit from comparing these algorithms with other recent detection methods to present a more comprehensive view of the current state of the art.

The methodology could be improved by including additional controls for variables affecting detection, such as varying lighting conditions, camera angles, and differences in cattle behavior. The study could also benefit from more advanced data augmentation techniques and exploring the impact of different hyperparameter settings. Implementing rigorous controls for experimental variability and a detailed analysis of potential sources of bias in the images could enhance the robustness of the results. Additionally, investigating other detection methods and conducting experiments under different conditions could offer a more thorough assessment.

The authors filmed the animals for only 7 minutes during the second feeding offer. It is unclear what criteria were used to select this second feeding and the duration of filming. The limited number of animals used may affect the study's replicability, as the animals might display biases towards certain behaviors, making the sample size not representative. Therefore, the observed number of animals is not considered significant.

The authors should consider a more detailed discussion of the evaluated metrics such as precision, recall, mAP, and F1-score, and compare these with other studies. A comprehensive analysis of how these metrics impact algorithm selection and practical applicability could enrich the work. Additionally, including additional controls for variables that may influence detection, such as lighting conditions and behavioral variations, is recommended. Applying advanced data augmentation techniques and conducting a thorough analysis of potential biases in the images would also help improve result robustness.

The authors do not sufficiently discuss the low metrics for the “Visiting” behavior and the potential practical implications.

The conclusions are consistent with the evidence presented, showing that YOLO v10 outperforms YOLO v8 in detecting specific feeding behaviors. The comparison of training times is well-supported by the data. However, the study would benefit from a more detailed analysis of why YOLO v10 performs better, particularly for the “Visiting” activity, where YOLO v10 showed better recall and F1-score. Additionally, a more in-depth discussion of cases where YOLO v8 performed better and how these differences impact practical applicability is needed. The authors should avoid making broad claims about “accuracy” when considering all metrics, especially given the low metrics for “Visiting.”

Author Response

The main issue addressed by the research is the effectiveness of YOLO v8 and v10 algorithms in detecting feeding behaviors in cattle from frontal images. The study aims to evaluate which version of the algorithm performs better in identifying specific activities such as “Chewing,” “Biting,” and “Visiting.” Although the research tackles a relevant practical application, the choice of these specific algorithms and the rationale for comparing only these versions could be more thoroughly justified. An analysis explaining why these algorithms were chosen over more recent ones or alternative technological approaches would strengthen the study.

Yes, a brief explanation about the choice of these specific algorithms was missing, we added some lines (52 to 56), to clarify why YOLO instead of the others. In the following comments we will address the question of explaining the three activities. However, according to the two different versions of YOLO evaluated here, we wrote a brief review about the YOLO algorithms since its beginning, and at the end, we explain the choice of the two versions that we utilized here, we consider that more explanation about the versions will be too much but we are open to discuss. Thanks.

Comments 2:

The authors also need to better justify the significance of each behavior in the study and its impact. The originality of the research lies in the practical application of YOLO algorithms for detecting specific feeding behaviors in cattle, a field that can significantly impact efficient herd management and health. The specific gap addressed is the lack of detailed comparison between the latest YOLO versions in the context of cattle monitoring. The study contributes by demonstrating how architectural improvements in YOLO v10 result in superior performance compared to YOLO v8. However, a deeper exploration of the relevance of these feeding behaviors to productivity and animal health, as well as a discussion on how automated detection can be integrated into existing management practices, would provide additional value.

Agree, this information was missing, we added some lines specifying why these activities and the possible implication and importance of determining them on-farm conditions. Lines were added both in the introduction to justify the evaluation of these 3 activities (Lines 79-85) and in the final conclusion (Lines 368-372).

Comments 3:

This study offers valuable insights by directly comparing YOLO v8 and v10 in the specific task of detecting feeding behaviors in cattle. It provides a detailed analysis of architectural improvements and their practical implications. While many studies focus on general performance comparisons, this work highlights practical and technical differences between algorithm versions, which can guide more informed choices for automated monitoring tools. Nevertheless, the study would benefit from comparing these algorithms with other recent detection methods to present a more comprehensive view of the current state of the art.

We understand the point here, we already added why we used YOLO instead of other objects detection algorithms, actually, YOLO represents the best trade-off between efficiency, accuracy and speed for real-time applications. We are interested in applying these technologies on-farm and in real-time, therefore, YOLO is the best option.

Comments 4:

The methodology could be improved by including additional controls for variables affecting detection, such as varying lighting conditions, camera angles, and differences in cattle behavior. The study could also benefit from more advanced data augmentation techniques and exploring the impact of different hyperparameter settings. Implementing rigorous controls for experimental variability and a detailed analysis of potential sources of bias in the images could enhance the robustness of the results. Additionally, investigating other detection methods and conducting experiments under different conditions could offer a more thorough assessment.

Due to a combination of logistical, ethical, and resource-based constraints, it was not feasible to capture additional images for our study. The resources allocated for this project had already been fully utilized, covering the costs associated with our existing dataset. Ethically, we were committed to ensuring the welfare of the cattle involved, and increasing the frequency of data collection could potentially have stressed these animals, conflicting with animal welfare standards. Moreover, operational schedules at the collaborating farms had to be respected, as these were active agricultural environments where disruptions could negatively impact daily operations. Seasonal changes also posed a challenge, as they could significantly affect cattle availability and behavior, making further data collection unfeasible outside the initial planned timeframe.

Comments 5:

The authors filmed the animals for only 7 minutes during the second feeding offer. It is unclear what criteria were used to select this second feeding and the duration of filming. The limited number of animals used may affect the study's replicability, as the animals might display biases towards certain behaviors, making the sample size not representative. Therefore, the observed number of animals is not considered significant.

We understand, and that is true, we decided to use the second meal because when we recorded in the first meal (the main one), animals were hungry and presented several dominances and behaviors that may bias our predictions. According to the 7 mins, animals were on average 7 mins in the feeder, after that they abandoned this place to go back to the pen and ruminate. We added some lines clarifying these facts (99 to 100 and 105 to 107).

Comments 6:

The authors should consider a more detailed discussion of the evaluated metrics such as precision, recall, mAP, and F1-score, and compare these with other studies. A comprehensive analysis of how these metrics impact algorithm selection and practical applicability could enrich the work. Additionally, including additional controls for variables that may influence detection, such as lighting conditions and behavioral variations, is recommended. Applying advanced data augmentation techniques and conducting a thorough analysis of potential biases in the images would also help improve result robustness.

We just added several lines explaining better all the evaluated metrics (Lines 380 to 425).

Comments 7:

The authors do not sufficiently discuss the low metrics for the “Visiting” behavior and the potential practical implications.

That is totally true, we did not explain visiting because it’s the less important activity but now we added some lines (283 to 289) explaining why we found much lower performances in visiting, thanks.

Comments 8:

The conclusions are consistent with the evidence presented, showing that YOLO v10 outperforms YOLO v8 in detecting specific feeding behaviors. The comparison of training times is well-supported by the data. However, the study would benefit from a more detailed analysis of why YOLO v10 performs better, particularly for the “Visiting” activity, where YOLO v10 showed better recall and F1-score. Additionally, a more in-depth discussion of cases where YOLO v8 performed better and how these differences impact practical applicability is needed. The authors should avoid making broad claims about “accuracy” when considering all metrics, especially given the low metrics for “Visiting.”

We just added several lines explaining better all the evaluated metrics (Lines 428 to 455).

Reviewer 2 Report

Comments and Suggestions for Authors

The manuscript addresses a topical aspect of PLF and the experimental research carried out by the authors was designed on a sound scientific basis. Moreover, the results provide meaningful insights about the application of computer vision for the monitoring and behavior detection and classification of cattle.

There are some significant aspects which should be improved by the author to make the manuscript suitable for publication.

First of all, the introduction section should be integrated with a more specific analysis of the scientific literature about the monitoring of feeding behavior udf cattle through computer vision. In particular, it is the case to underline why the detection of eating, chewing, and visiting behavior is important and for what purpose: what is the usefulness of such data for the development of analyses concerning animal welfare, health, or productivity . This information is fundamental to properly calibrate the computer vision approach, in a way that can provide the most relevant data for such analyses.

Another important aspect which has not been adequately addressed in the manuscript regards the identification of the animals. The method proposed by the authors is effective in detecting and classifying specific actions at the feeding line. Still, no indication is provided about how the data collected can be linked with the respective bull. It is well known that associating data with individual animals is a key point of PLF.

Then, a discussion section is missing and the conlusions are very sinthetc and poorly informative: please provide a thorough discussion of the results, with reference to the issues pointed out in the (revised) introduction.

Specific comments

Lines 88-90: Please rewrite the sentence in a way that clearly explains the overall length of the video recorded and the time interval covered by video recording.

Lines 112-113: Please indicate the total number of frames considered.

Subsection 2.3: Please report, for each of the 6 bulls, the number of frames where it appears

Results section: Please report also the precision recall diagrams, which are useful to increase the effectiveness of the representation of the performances of the deep learning networks. 

Comments on the Quality of English Language

The text is generally correct, but it would benefit from a thorough revision with the support of a native speaker, to improve the overall clearness of the presentation.

Author Response

Response to Reviewer 2 Comments

The manuscript addresses a topical aspect of PLF and the experimental research carried out by the authors was designed on a sound scientific basis. Moreover, the results provide meaningful insights about the application of computer vision for the monitoring and behavior detection and classification of cattle.

Thanks for these words, we will try to address as better as possible your following comments.

Comments 1:

There are some significant aspects which should be improved by the author to make the manuscript suitable for publication.

Okay, lets go for it, again, thanks.

First of all, the introduction section should be integrated with a more specific analysis of the scientific literature about the monitoring of feeding behavior udf cattle through computer vision. In particular, it is the case to underline why the detection of eating, chewing, and visiting behavior is important and for what purpose: what is the usefulness of such data for the development of analyses concerning animal welfare, health, or productivity. This information is fundamental to properly calibrate the computer vision approach, in a way that can provide the most relevant data for such analyses.

Yes, that is true, both reviewers coincided in this point so I added some lines explaining why the interest of these specific activities (Lines 79 to 85), I hope this may clarify this misunderstanding.

Comments 2:

Another important aspect which has not been adequately addressed in the manuscript regards the identification of the animals. The method proposed by the authors is effective in detecting and classifying specific actions at the feeding line. Still, no indication is provided about how the data collected can be linked with the respective bull. It is well known that associating data with individual animals is a key point of PLF.

Okay, this is a very good point, thanks, we may see that you already worked with computer vision. Well, in our case we tried to detect animal’s identification through an RFID reader placed in the feeder, however, this system did not work properly and we had to do it manually. Therefore, during the experiment and the recording, we wrote down the animal’s position from left to right with their respective identification numbers. To clarify this point we added some lines (Lines 115 to 116).

Comments 3:

Then, a discussion section is missing and the conclusions are very synthetic and poorly informative: please provide a thorough discussion of the results, with reference to the issues pointed out in the (revised) introduction.

True, we added a discussion section (Lines 367 to 405) in order to specifically address the differences in prediction performance across activities and possible ways to improve these parameters.

Specific comments:

Lines 88-90: Please rewrite the sentence in a way that clearly explains the overall length of the video recorded and the time interval covered by video recording.

Done, we added some lines to clarify (lines 97 to 100).

Lines 112-113: Please indicate the total number of frames considered.

Done

Subsection 2.3: Please report, for each of the 12 bulls, the number of frames where it appears

The distribution of the bulls across the videos reveals that 41.7% of the bulls (5 out of 12) appear in three or fewer videos, indicating a lower frequency of appearance. Meanwhile, 33.3% of the bulls (4 out of 12) are present in 11 to 13 videos, representing a moderate level of occurrence. Lastly, 25% of the bulls (3 out of 12) appear in 16 or more videos, signifying a higher frequency of presence. This distribution highlights the varying degrees of visibility for each bull across the analyzed frames set.

Results section: Please report also the precision recall diagrams, which are useful to increase the effectiveness of the representation of the performances of the deep learning networks.

Precision-Recall diagrams reported at the end of the manuscript (Lines 398 to 399).

Comments on the Quality of English Language:

The text is generally correct, but it would benefit from a thorough revision with the support of a native speaker, to improve the overall clearness of the presentation.

Thanks, we will try to review it by a native English speaker.